# Subtype-selective agonists of plant hormone co-receptor COI1-JAZs identified from the stereoisomers of coronatine

Kengo Hayashi[1], Nobuki Kato[1], Khurram Bashir[2,3], Haruna Nomoto[1], Misuzu Nakayama[1], Andrea Chini [4], Satoshi Takahashi[2], Hiroaki Saito[5], Raku Watanabe[6], Yousuke Takaoka[1], Maho Tanaka[2], Atsushi J. Nagano [7,8], Motoaki Seki [2], Roberto Solano [4] & Minoru Ueda [1,6✉]

Severe genetic redundancy is particularly clear in gene families encoding plant hormone receptors, each subtype sharing redundant and specific functions. Genetic redundancy of receptor family members represents a major challenge for the functional dissection of each receptor subtype. A paradigmatic example is the perception of the hormone (+)-7-*iso*-jasmonoyl-L-isoleucine, perceived by several COI1-JAZ complexes; the specific role of each receptor subtype still remains elusive. Subtype-selective agonists of the receptor are valuable tools for analyzing the responses regulated by individual receptor subtypes. We constructed a stereoisomer library consisting of all stereochemical isomers of coronatine (COR), a mimic of the plant hormone (+)-7-*iso*-jasmonoyl-L-isoleucine, to identify subtype-selective agonists for COI1-JAZ co-receptors in *Arabidopsis thaliana* and *Solanum lycopersicum*. An agonist selective for the *Arabidopsis* COI1-JAZ9 co-receptor efficiently revealed that JAZ9 is not involved in most of the gene downregulation caused by COR, and the degradation of JAZ9-induced defense without inhibiting growth.

[1] Department of Chemistry, Graduate School of Science, Tohoku University, Sendai 980-8578, Japan. [2] Plant Genomic Network Research Team, RIKEN Center for Sustainable Resource Science, Yokohama 230-0045, Japan. [3] Department of Life Sciences, SBA School of Science and Engineering, Lahore University of Management Sciences, 54792 Lahore, Pakistan. [4] Plant Molecular Genetics Department, National Centre for Biotechnology (CNB), Consejo Superior de Investigaciones Cientificas (CSIC), Campus University Autonoma, 28049 Madrid, Spain. [5] Faculty of Pharmaceutical Sciences, Hokuriku University, Kanazawa 920-1181, Japan. [6] Department of Molecular and Chemical Life Sciences, Graduate School of Life Sciences, Tohoku University, Sendai 980-8578, Japan. [7] Faculty of Agriculture, Ryukoku University, Shiga 520-2194, Japan. [8] Institute for Advanced Biosciences, Keio University, Yamagata 997-0017, Japan. ✉email: minoru.ueda.d2@tohoku.ac.jp

Sub-functionalization or neo-functionalization of genes occurs during evolution through duplication events in the genome. Gene duplications leading to genetic redundancy can allow the acquisition of new functions, interactions, and expression patterns to benefit plant adaptive responses to the changing environmental stimuli[1–3]. Examples of severe genetic redundancy can be observed in plant hormone signaling modules[1,2]. Genetic redundancy in plant hormone receptors makes it difficult to study and characterize their complex signaling networks[3]. A paradigmatic example can be found in the case of a lipid-derived plant hormone, (+)-7-*iso*-jasmonoyl-L-isoleucine (JA-Ile, **1**, Fig. 1a), which is perceived by COI1-JAZ co-receptors consisted of F-box protein CORONATINE INSENSITIVE 1 (COI1) and JASMONATE-JIM DOMAIN (JAZ) repressor proteins. Subsequent degradation of JAZ through the 26S-proteasome induces the expression of jasmonate (JA)-responsive genes, including defenses against necrotrophic pathogens and herbivorous insects in *Arabidopsis thaliana* and several crop plants[4–6]. Concurrently, the COI1-JAZ-mediated JA signaling regulates growth inhibition, fertility, and secondary metabolite production. The molecular basis of the multiple effects of JA-Ile partially lies in the severe genetic redundancy of the 13 *JAZ* gene family members encoded in the *Arabidopsis* genome[3,7,8].

Differences in spatiotemporal expression of *JAZ* genes during growth and development lead to redundant and complementary JA signaling in *Arabidopsis*. Genetic redundancy seriously challenges canonical genetic approaches to functionally dissect *JAZ* gene function as the effect of a single *JAZ* knock-out in *Arabidopsis* can be easily complemented by other members of the *JAZ*

gene family[3]. Tremendous efforts have been made to dissect the redundant function of *JAZ* genes in jasmonate signaling. Genetic studies using *Arabidopsis* mutant lines lacking multiple *JAZ* genes (*jaz1/3/4/9/10* quintuple mutant as *jazQ*, *jaz1/2/3/4/5/6/7/9/10/13* decuple mutant as *jazD*, *jaz1/2/3/4/5/6/7/8/9/10/13* as *jazU*, and recently reported *jaz1/2/3/4/5/6/7/9/10/11/12* undecuple mutant) are pioneering works to understand the function of JAZ family genes[9–12]. Recently, genetic studies using the model liverwort *Marchantia polymorpha* L have attracted attention as an alternative approach to analyze the jasmonate signaling because no genetic redundancy is observed in the key JA signaling pathway components, COI1, JAZ, and MYC[13–17]. In addition, chemical genetics is a powerful and promising alternative. Chemical genetics approaches have been successful in analyzing genetically redundant signaling in other hormonal pathways, such as those of abscisic acid and strigolactones, in which specific hormonal agonists activated distinctive receptor subtypes triggering specific responses[18–24]. Transient inhibition/degradation of a single JAZ protein in a fully developed plant by using chemicals may therefore disclose the specific function of a particular *JAZ* gene. Thus, the chemical genetic approach can complement the genetic approach[25,26].

Here we report that the analysis of the chemical library (Fig. 1b) harboring all the stable stereochemical isomers of JA-Ile mimic, coronatine (COR, **2**, Fig. 1a), provided the agonists of selective affinity among the COI1-JAZ co-receptor pairs in two plant species, *A. thaliana* and *Solanum lycopersicum*. Significantly, the stereoisomer **2e** (Fig. 1b) caused protein degradation of the JAZ9 repressor to activate JAZ9-dependent jasmonate signaling in *A. thaliana*. The use of **2e** enabled the analysis of a unique gene expression pattern governed by a single JAZ repressor, JAZ9, among the genetically redundant JAZ family genes.

## Results

### Chemical synthesis of all possible stereochemical isomers of coronatine.

The use of a focused chemical library of **2**-stereoisomers (Fig. 1b) can be a promising approach to look for novel, specific ligands of COI1-JAZ co-receptor pairs. Stereochemical isomers of **2** have the same size exclusion volume as naturally occurring **2** and slightly different 3D orientation of functional groups in a molecule, thus they can be accommodated in the ligand-binding pocket of COI1 with modulation of interaction network to each co-receptor pair leading to selectivity among JAZs (Fig. 1c).

The chemical library of all the stable **2**-stereoisomers was constructed by coupling stereoisomers of coronafacic acid (CFA; **3**) and coronamic acid (CMA; **4**) (Fig. 1a). Enough amount of the stable stereoisomers of **3** was achieved by gram-scale racemic CFA synthesis and subsequent optical resolution, epimerization, and separation of diastereomers (Supplementary Fig. 1a)[27–29]. The optical purities of four CFAs (**3**-**3c**) were determined by chiral HPLC analyses[30]. Four CFAs (**3**-**3c**) were coupled with four CMAs (**4**-**4c**) which were synthesized by the reported method[31], and subsequent deprotection provided all the stable 16 stereochemical isomers of coronatine (**2** and **2a**-**2o** in Fig. 1a, b, and Supplementary Figs. 1b and 2).

### Stereochemical isomers demonstrated unique selectivity for COI1-JAZ co-receptor pairs of two plant species, *Arabidopsis thaliana* and *Solanum lycopersicum*.

We examined the affinities of 16 stereochemical isomers of **2** with COI1-JAZ co-receptors of two plant species, *Arabidopsis thaliana* (encoding 13 JAZ proteins), as a representative model dicot, and *Solanum lycopersicum*

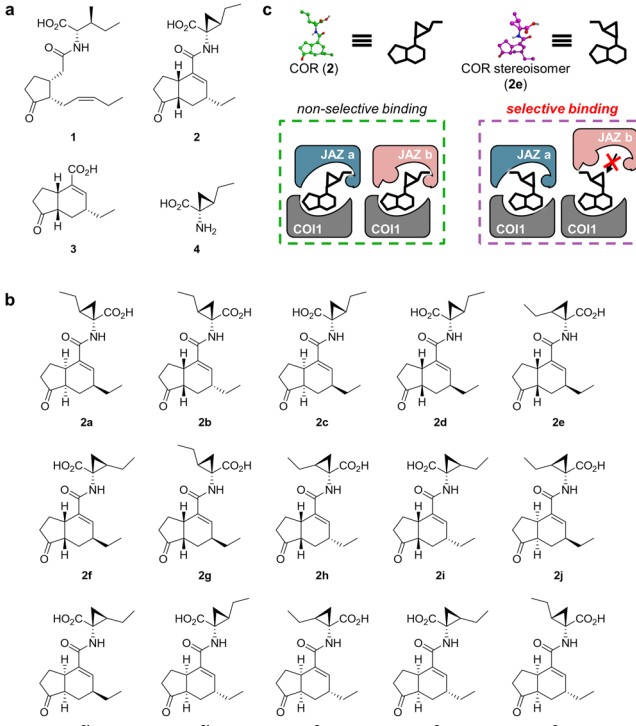

**Fig. 1 Chemical library of coronatine stereoisomers. a** Chemical structures of JA-Ile (**1**), coronatine (**2**), coronafacic acid (**3**), and coronamic acid (**4**). **b** Chemical structure of stereochemical isomers of **2** (**2a**-**2o**). **c** Schematic representation of selective binding of **2**-stereoisomer (**2e**) with COI1-JAZ co-receptors. COR (**2**) can interact with most of the COI1-JAZ pairs, whereas the stereoisomer of COR (**2e**) has a selective affinity with some COI1-JAZ pairs.

(encoding 13 JAZ proteins), as a model plant belonging to the *Solanaceae* family (Supplementary Fig. 3). Chemical screening using in vitro fluorescence anisotropy (FA) assay was performed for all of the functional combinations of COI1-JAZ co-receptor pairs[8,32].

For *Arabidopsis* COI1-JAZ pairs, in vitro FA assay was carried out for all of the functional combinations of *At*COI1-*At*JAZ1/2/3/4/5/6/9/10/11/12 (Supplementary Fig. 3)[33]. *At*COI1-*At*JAZ7/8/13 pairs were excluded in all of the following assays because *At*JAZ7/8/13 lack the short conserved canonical degron sequence and thus cannot form the *At*COI1-**1/2**-*At*JAZ complex[34,35]. Instead of full-length *At*JAZ proteins, we used the fluorescein-labeled *At*JAZ degron short peptides (Fl-*At*JAZPs), including the ligand-interacting domain of full-length *At*JAZ proteins[33,36]. FA assay provided the binding affinity ($K_d$) of each stereochemical isomer of **2** against *At*COI1-*At*JAZ pairs (Fig. 2a, and Supplementary Figs. 4 and 5). The results of the FA assay were summarized as an affinity 'heatmap' profile of 16 stereochemical isomers (Fig. 2a). Based on their affinity, stereochemical isomers were categorized into four groups; no affinity (**2a**, **2b**, **2g**, **2j**, **2k**, **2m**, **2n**, and **2o**), non-selective affinity (**2**, **2d**, **2h**, **2i**, **2l**), moderately selective affinity (**2c**), and highly selective affinity (**2e**, **2f**) to some of the *At*COI1-*At*JAZ combinations (Fig. 2a and Supplementary Fig. 5). In vitro pull-down assays using Fl-*At*JAZP and GST-*At*COI1 protein confirmed the highly selective affinity of **2e** and **2f** (Supplementary Fig. 6). Surprisingly, **2e** has a selective affinity to the *At*COI1-*At*JAZ9 pair in FA and pull-down assays. Docking and subsequent in silico molecular dynamics (MD) simulations demonstrated that **2e**, as well as **2**, can be accommodated in the binding pocket of COI1-InsP$_8$-JAZ9 complex (Supplementary Fig. 7). In this MD simulation, we used inositol octakisphosphate (InsP$_8$) instead of inositol pentakisphosphate (InsP$_5$) because previous reports showed that InsP$_8$ also formed stable complex with COI1-**2**-JAZ instead of phosphates (PO$_4$)[37,38]. In planta *At*JAZ9-selectivity of **2e** was tested by using a GUS reporter assay using *A. thaliana* reporter lines, *35S: JAZ1-GUS*, *35S: JAZ9-GUS*, *35S: JAZ10-GUS*, and *35S: JAZ12-GUS* (Fig. 2b and Supplementary Fig. 8). All JAZ-GUS fusion proteins were degraded in **2**-treated seedlings. In contrast, **2e** caused protein degradation in the *At*JAZ9-GUS reporter line, whereas no degradation was observed in the *At*JAZ1-GUS and *At*JAZ10-GUS reporter lines. In addition, **2e**-treatment caused degradation of JAZ9-GFP protein in *35S: JAZ9-GFP*, but not of JAZ1-GFP protein in *35S: JAZ1-GFP* (Supplementary Fig. 9). These results confirmed that **2e**-treatment caused the selective 'protein knock-out (degradation)' of JAZ9 protein in *A. thaliana*.

For *Sl*COI1-*Sl*JAZ pairs of *S. lycopersicum*, in vitro FA assay was also carried out for *Sl*COI1-*Sl*JAZ1-11/13 (Fig. 2c). Phylogenetic characterization and functional annotation of *Sl*JAZs were reported previously (Supplementary Fig. 3)[32]. Based on this result, we reported the affinity between **1/2** and *Sl*COI1-*Sl*JAZ pairs using GST-*Sl*COI1 and fluorescein-labeled *Sl*JAZ degron short peptides (Fl-*Sl*JAZPs)[39]. According to this previous report, *Sl*JAZ12 was excluded in the FA assay because it does not have a conserved JAZ degron sequence[32]. The stereochemical library showed distinct selectivity against *Sl*COI1-*Sl*JAZ pairs (Fig. 2c and Supplementary Figs. 10 and 11), as with the case of *At*COI1-*At*JAZ pairs. Among them, remarkable selectivity was found for the stereochemical isomers **2e** and **2f**. **2e** has selectivity against *Sl*COI1-*Sl*JAZ5/6/8, and **2f** has selectivity against *Sl*COI1-*Sl*JAZ5/6/8/13. Unfortunately, we cannot examine the affinity of stereoisomers with the *Sl*COI1-*Sl*JAZ7P pair in the FA assay because the severe non-specific binding was observed in the absence of ligand. Interestingly, isomers **2e** and **2f** also showed selectivity against *At*COI1-*At*JAZ pairs (Fig. 2a).

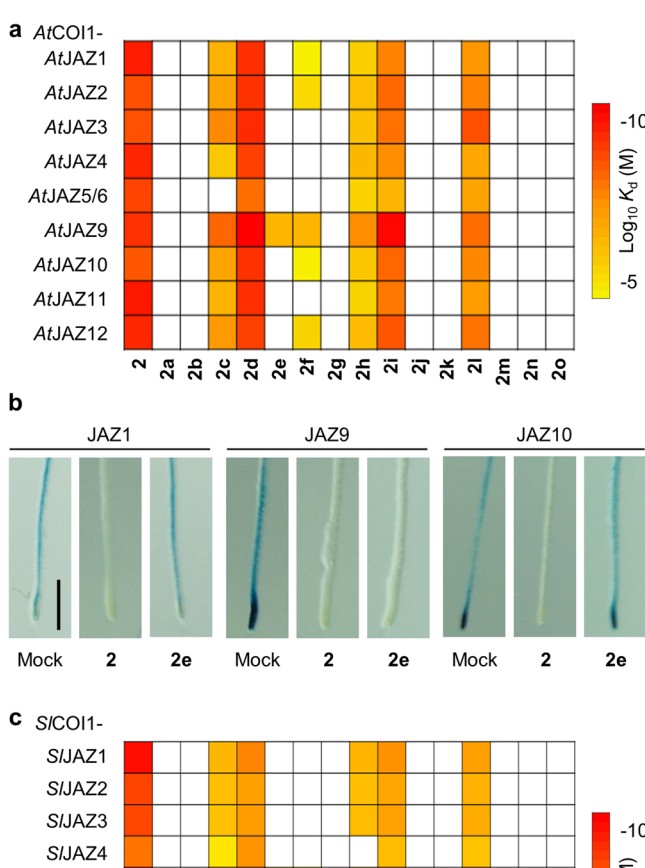

**Fig. 2 Affinity heatmaps of coronatine stereoisomers on COI1-JAZ co-receptors of *A. thaliana* and *S. lycopersicum*. a** Heatmap analyses of $K_d$ value of all stereoisomers for functional *At*COI1-*At*JAZ pairs. White means "+not determined" due to a low increase of FA (Supplementary Fig. 4). The color bar represents the $K_d$ value of each stereoisomer. $K_d$ value of each stereoisomer which increased FA value at 1 μM was calculated by dose-response. **b** Evaluation of GUS activity in the roots of 4-day-old *35S: JAZ1-GUS*, *35S-JAZ9-GUS*, and *35S: JAZ10-GUS* seedlings. Images of each seedling stained with 5-bromo-4-chloro-3-indolyl glucuronide were pretreated for 2 h with or without ligand (1 μM). Similar results were obtained in three independent experiments. Scale bar, 1 mm. **c** Heatmap analyses of $K_d$ values of all stereoisomers for *Sl*COI1-*Sl*JAZ pairs. White means "not determined" due to a low increase of FA (Supplementary Fig. 10). The color bar represents the $K_d$ value of each stereoisomer. $K_d$ value of each stereoisomer which increased FA value at 1 μM was calculated by dose-response curves.

Through these examinations, we found two stereochemical isomers, **2e** and **2f**, showing distinct selectivity for COI1-JAZ co-receptor pairs of two plant species. These results demonstrated that the stereochemical library of **2** is a promising chemical tool for discovering ligands with selective affinity for some subtypes of the genetically redundant COI-JAZ co-receptor family in higher plants.

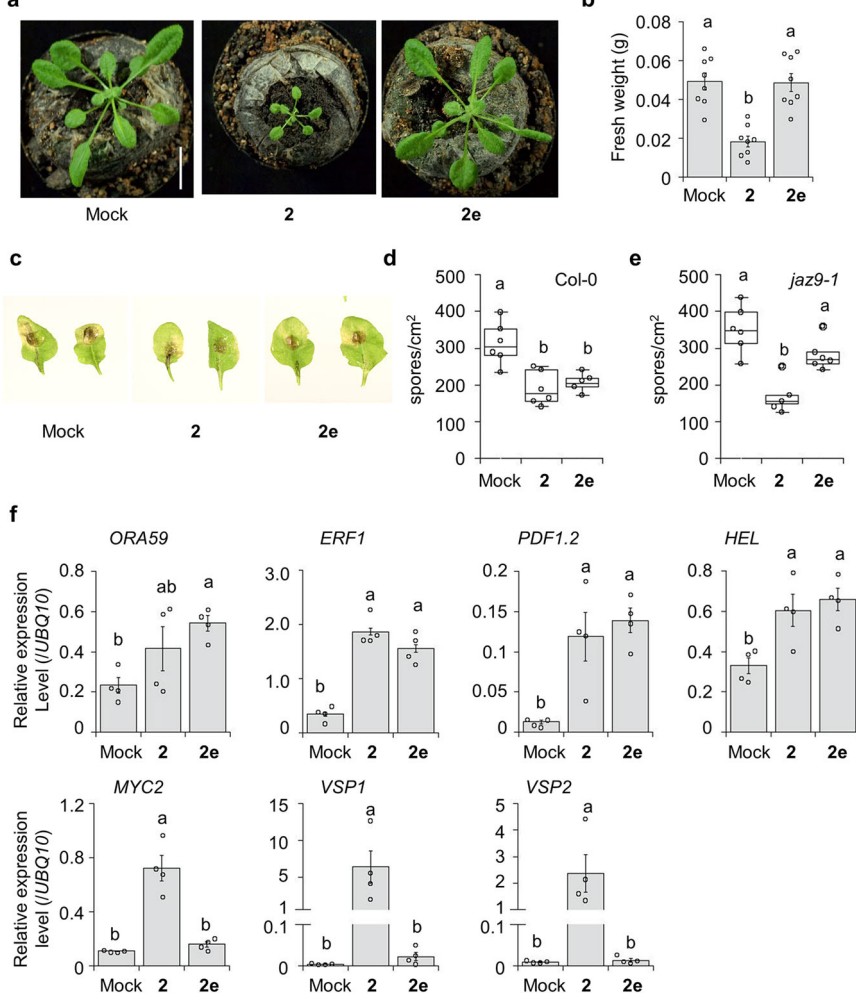

**Fig. 3 In planta analyses of stereoisomers in *Arabidopsis thaliana*. a, b** Growth inhibition assay using ligand-treated 4-week-old *A. thaliana*.
**a** Representative images of the **2**- (50 μM) and **2e**-treated (50 μM) plants. Similar results were obtained in three independent experiments (Supplementary Fig. 12). Scale bar: 1.0 cm. **b** Quantification of fresh weight of aerial part of the ligand-treated plants (*n* = 8). Significant differences were evaluated by one-way ANOVA/Tukey HSD post hoc test (*p* < 0.05). Values are mean ± SE. **c–e** Fungal infection assay using **c**, **d** 4-week-old Col-0 and **e** 4-week-old *jaz9-1*. Plants were treated with mock solution, **2** or **2e** (50 μM) and infected with *A. brassicicola* (*n* = 7). **c** Representative images of leaves infected with *A. brassicicola* are shown in (**d**) and (**e**). Quantification of fungal spores was undertaken 5 days after inoculation; the results are depicted using box plots; horizontal lines are medians, boxes show the interquartile range, and error bars show the entire data range. Significant differences were evaluated by one-way ANOVA/Tukey HSD post hoc test (*p* < 0.05). Experiments were repeated three times with similar results. Values are mean ± SE. **f** Gene expression analysis in 4-week-old *A. thaliana* with or without compound treatment (50 μM, *n* = 4) without pathogen inoculation. Each sample was flash-frozen after 8 h from definitive treatment. Similar results were obtained in three independent experiments. Significant differences were evaluated by one-way ANOVA/Tukey HSD post hoc test (*p* < 0.05). Values are mean ± SE.

## A stereochemical isomer induced defenses without growth inhibition through selective activation of jasmonate signaling in *A. thaliana*.

To examine the power of stereochemical isomers to dissect the jasmonate signaling pathway, we focused on **2e**, with selective affinity to the *At*COI1-*At*JAZ9 pair (Fig. 2a). We explored the biological effect of **2e** and its mode of action on *A. thaliana*.

**2e** showed no growth inhibition on 4-week-old *A. thaliana*, whereas **2** caused severe growth inhibition (Fig. 3a, b, and Supplementary Fig. 12). In addition, no growth inhibition was observed on root by the addition of **2e** (Supplementary Fig. 13). However, **2e**, as well as **2**, moderately activated resistance response against the infection of fungal pathogen *Alternaria brassicicola* compared to the mock-treatment (Fig. 3c, d, and Supplementary Fig. 14)[36]. This **2e**-mediated defense trait was impaired in *jaz9-1*, in which other members of the JAZ family protein likely complemented the original function of JAZ9

protein in presence of non-selective ligand, and **2e** cannot cause their degradation (Fig. 3e and Supplementary Fig. 15) This result was in clear contrast to **2**, which caused a close defense trait in both Col-0 and *jaz9-1* (Fig. 3c–e and Supplementary Fig. 15). These effects of **2e** on *A. thaliana* strongly suggested that **2e** can induce a resistance response by activating the *At*JAZ9-mediated partial jasmonate signaling. Reverse transcription-qPCR (RT-qPCR) analyses demonstrated that **2e**-treatment in 4-week-old *A. thaliana* (without pathogen infection) caused the upregulation of defense-related genes, such as *ORA59*, *ERF1*, *PDF1.2*, and *HEL*, without affecting the expression of JA marker genes, such as *MYC2* and *VSP1/2* (Fig. 3f). In the downstream of JA signaling, *ORA59, ERF1*, *PDF1.2*, and *HEL* belong to ERF/ORA branch, and *MYC2* and *VSP1/2* belong to the MYC2-branch (Supplementary Fig. 16). Thus, **2e** selectively upregulated the ERF/ORA branch of JA-signaling without affecting MYC-branch. In addition, **2e**-mediated expression of *ORA59* and *PDF1.2* was impaired in *coi1-*

**1**, which confirmed the COI1-dependent mode-of-action of **2e** (Supplementary Fig. 17).

**Chemical and genetic AtJAZ9 knock-out showed distinct gene expression profiles.** Previous reverse genetic studies demonstrated that *jaz9-1* has no significant phenotype regarding root-growth inhibition and defense response[10,11]. In contrast, our **2e**-induced protein knock-out of JAZ9 caused activation of the defense trait. To examine the distinct difference between chemically induced protein degradation or loss-of-function of JAZ9 by **2e** and genetic knock-out in *jaz9-1*, RNA sequencing (RNAseq) analyses were carried out for a comprehensive comparison of gene expression profiles between 6-day-old seedlings of WT (Col-0) treated by **2/2e** and *jaz9* mutant (*jaz9-1*) treated by mock (Supplementary Data 1).

To analyze the entire gene expression profiles, we performed principal component analysis, Venn diagram analyses, and heatmap analyses of upregulated genes with >2.5-fold changes (Fig. 4a–c). In principal component analysis, no significant difference was observed between gene expression profiles of the mock-treated *jaz9-1* and the mock-treated Col-0 (Fig. 4a), suggesting that in the *jaz9-1* mutant, most functions of *JAZ9* as a transcriptional repressor would be complemented by other partially redundant members of the *JAZ* family. In contrast, a clear difference was found between the gene expression profiles of **2e**-treated Col-0 and mock-treated *jaz9-1* (Fig. 4a). This result showed a distinct difference in gene expression profiles between genetic knock-out and chemically induced protein degradation (or loss-of-function).

We further examined the unique gene expression caused by **2e**. Detailed analyses of the gene expression profiles shown in the Venn diagram and heatmap also showed that **2e** caused a part of **2**-mediated gene expression: **2** upregulated expressions of 950 genes, of which **2e** upregulated 103 genes (Fig. 4b). A significant difference was observed between the results of **2e**-induced chemical degradation (or knock-out) and genetic knock-out of *JAZ9*: 91 of 103 gene upregulations caused by **2e** were not observed in *jaz9-1* (Fig. 4b and Supplementary Fig. 18 as a list of 91 genes). Heatmap analysis (Fig. 4c) showed that **2** upregulated major JA signaling-related genes, such as *JAZ*, JA-responsive genes, defense-related genes, and metabolism-related genes, whereas **2e** primarily upregulated expressions of ethylene-mediated defense-related genes (see also Supplementary Figs. 19 and 20).

To our surprise, a remarkable difference in expression profiles was observed in downregulated genes: **2e** caused downregulation of gene expression only for eight genes, whereas **2** caused downregulation for 614 genes (Fig. 4d and Supplementary Fig. 19). GO analysis of **2**-repressed genes showed that GO terms related to cell wall organization/biogenesis were significantly repressed (Fig. 4d). RT-qPCR analyses showed that **2** repressed the gene expression of cell wall organization/biogenesis-related genes such as *EXLA2*, *XTH19*, and *XTH23*, which were previously reported as essential genes for plant root growth[40,41], in a *COI1*-dependent manner, but **2e** did not affect their expression (Fig. 4e). The reason why **2e** did not cause growth inhibition could be explained by the lack of gene downregulation of these cell-wall-related genes.

**A stereoisomer caused AtJAZ9-dependent gene expressions.** The fact that mode of action of **2e** is dependent mainly on JAZ9 was confirmed by a comparison of RNAseq data on **2e**-treated Col-0 and **2e**-treated *jaz9-1* (Fig. 4f). In *jaz9-1*, **2e** activated the expression of 100 genes, and 52 of 100 genes were upregulated in **2e**-treated Col-0, of which three were upregulated in mock-

treated *jaz9-1* (Fig. 4f). Thus, expressions of 49 genes (Supplementary Fig. 20a as a list of 49 genes) appear to be induced independently of *JAZ9* in the **2e**-treated *jaz9-1*. The 49 genes include JA-marker genes such as *JAZ1*, *MYC2*, and *AOS*, and defense-related genes belonging to the ERF/ORA branch, such as *ORA59*, *ERF1*, *PDF1.3*, and *PDF1.2* (Supplementary Fig. 21a). However, the RT-qPCR analysis showed that the expression levels of most of the 49 genes found in **2e**-treated *jaz9-1* were not statistically significant compared to those in **2e**-treated Col-0, while the expression levels of ERF/ORA-genes which are upregulated by **2e** in Col-0 were significantly suppressed in **2e**-treated *jaz9-1* (Fig. 4g and Supplementary Fig. 21b). These results showed that **2e** upregulated the expression of defense-related genes in a mostly JAZ9-dependent manner. This result agrees with the result in Fig. 3e, in which **2e** caused no remarkable resistance response against the pathogen in the *jaz9-1* mutant (Fig. 3e and Supplementary Fig. 15). To test if **2e** activity may also require JAZ10, as previously described for the JA-Ile selective agonist *N*-coronafacyl *ent*-coronamic acid *O*-phenyloxime (NOPh)[36], the requirement for JAZ10 in **2e**-induced responses was analyzed. **2e** did not affect the growth and upregulated defense-related genes belonging to the ERF/ORA branch in the *jaz10-1* mutant (Supplementary Fig. 22). Moreover, no growth inhibition and expressions of marker genes belonging to ERF/ORA branch were observed in **2e**-treated *jaz12-1* (Supplementary Fig. 23). These results rule out the role of JAZ10 in **2e**-mediated regulation and they suggested that **2e** upregulated the gene expressions mostly in a JAZ9-dependent manner.

We successfully developed the chemical tool **2e** which mediates JAZ9-selective degradation in planta. This **2e**-mediated activation leads to the expression of genes normally repressed by JAZ9 in *A. thaliana*. Examination of differential gene expression induced by JAZ9 protein degradation (or loss-of-function) revealed the unique function of JAZ9, which did not affect the downregulation of genes, including cell-wall-related ones which play important roles in the plant root growth, and caused upregulation of defense-related genes through the EIN3/EIL1-ERF/ORA59 pathway[40–43]. The results highlight the power of a stereochemical library of **2** to screen the JAZ-selective activator of jasmonate signaling in other plant species.

## Discussion

The chemical genetic approach is a promising way to dissect the plant hormone signaling of high genetic redundancy[25,26]. Previous studies have reported success in abscisic acid and strigo-lactone using small molecules screened from chemical libraries to elucidate the function of receptor subtypes belonging to genetically redundant receptor families[18,23,24]. Chemical libraries composed of planar molecules have been successfully used to screen hit compounds in drug discovery and plant chemical biology[26,44,45]. Chini et al. recently discovered a general antagonist of *At*COI1-*At*JAZ using a chemical library[46]. To the best of our knowledge, this was the only successful example of the chemical library screening of COI1-JAZ co-receptor agonist/antagonist. By contrast, we have screened a targeted library, based on stereoisomers of **2**, for highly subtype-selective ligands of 10 *At*COI1-*At*JAZ and 11 *Sl*COI1-*Sl*JAZ co-receptor pairs. **2e/2f**, in addition to high selectivity for *At*JAZ9, also showed selective affinity with *Sl*COI1-*Sl*JAZ5/6/8, which are the closest tomato homologs of *At*JAZ9 belonging to Group V (Supplementary Fig. 3). The chemical biology community focused on the conventional chemical libraries being composed of planar molecules, whereas we envisioned a chemical library composed of the stereoisomers of three-dimensional structures. These two approaches are complementary, and we demonstrated that

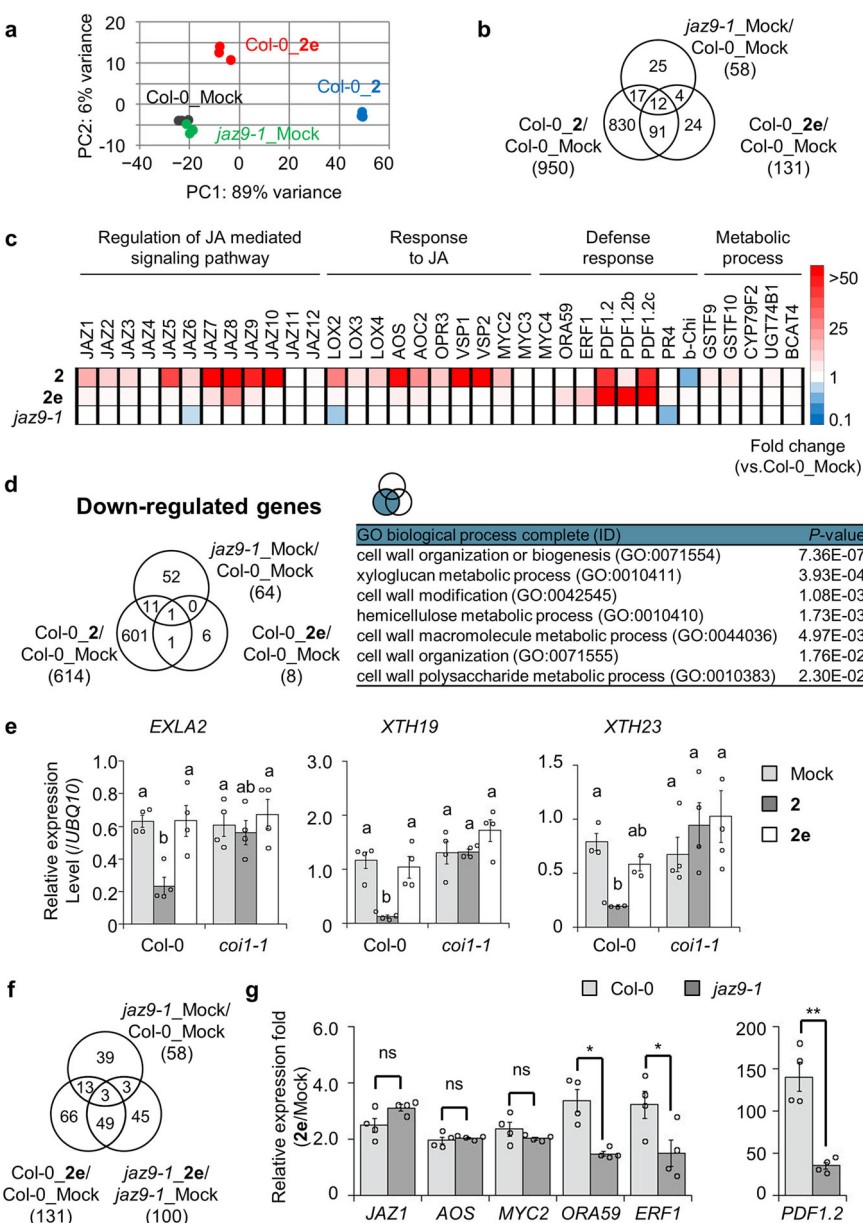

**Fig. 4 Gene expression analyses of 2- and 2e-treated Col-0 and jaz9-1 seedlings. a** Principal component analysis of upregulated genes in each compound-treated 6-day-old *Arabidopsis* seedlings by RNAseq analyses (fold change >2.5, FDR < 0.1). Black, blue, red, and green dots showed three independent replicates of each treatment. **b** Venn diagram (FDR < 0.1) indicates the number of upregulated genes at least 2.5-fold change in response to different treatments. **c** Heatmap illustrating changes in gene expression of jasmonate response to different treatments (FDR < 0.1). Each GO ID is as follows: Regulation of JA mediated signaling pathway: 2000022, Response to JA: 0009753, Defense response: 0006952, Metabolic process: 0008152. **d** Venn diagram (FDR < 0.1) indicating the number of downregulated genes at least 2.5-fold change in response to different treatments and GO enrichment analysis. **e** Gene expression analyses of cell wall organization-related genes by RT-qPCR in Col-0 and *coi1-1* seedlings (*n* = 4) with or without compound treatment (1 μM) for 8 h. Similar results were obtained in three independent experiments. Significant differences were evaluated by one-way ANOVA/ Tukey HSD post hoc test (*p* < 0.05). Values are mean ± SE. **f** Venn diagram (FDR < 0.1) indicating the number of upregulated genes at least 2.5-fold change in response to different treatments in Col-0 and *jaz9-1*. **g** RT-qPCR analyses in Col-0 and *jaz9-1* seedling (*n* = 4) with or without compound treatment (1 μM). Each relative expression fold was calculated using an average expression level in Mock- or **2e**-treated Col-0 or *jaz9-1* seedlings. Similar results were obtained in three independent experiments. Significant differences were evaluated by Student's t-test (**p* < 0.01, *p* < 0.05; ns, not significant). Values are mean ± SE.

stereochemical libraries could provide hit compounds that cannot be obtained from a chemical library of planar molecules.

In the present study, we have successfully identified *At*COI1-*At*JAZ9-selective agonist **2e** from the stereochemical library of **2**. Taking advantage of the JAZ9-selective mode-of-action of **2e**, we successfully find that JAZ9 plays a unique function in jasmonate signaling: JAZ9 does not participate in the gene downregulation

caused by **2**. Among the genes specifically repressed by **2**, but not by **2e**, we focused on genes related to cell wall formation and biosynthesis, such as *EXLA2*, *XTH19*, and *XTH23* (Fig. 4e). Previous reports suggested that these genes play important roles in root growth: overexpression lines of *EXLA2* and double knock-out mutant of *XTH19* and *XTH23*, which affect the root growth and lateral root formation[40,41,47]. Thus, the current results answered

the question, why **2e** did not affect the growth inhibition (Fig. 3a, b). In addition, using **2e**, we demonstrated that pathogen-resistance through ERF/ORA-branch JA-signaling is controlled by JAZ9-meditated jasmonate signaling. These findings agree with our previous results, in which NOPh has a selective affinity with *AtCOI1-AtJAZ9/10* pairs to enhance resistance response against pathogen *A. brassicicola* without causing growth inhibition[36]. NOPh selectively activates ERF/ORA-branch to unpair growth and defense through COI1/JAZ9-EIN3/EIL1-ERF/ORA59 signaling pathway[42]. In contrast to NOPh, **2e** mostly caused the selective degradation of single JAZ repressor, thus enabling analysis of gene expression profiles regulated by a single JAZ. **2e** is specific mainly for JAZ9, although RNAseq suggest that it might target other JAZ at least partially and weakly. However, since the **2e**-mediated expression level of *JAZ9*-independent JA-responsive genes is low (Fig. 4g and Supplementary Fig. 21b), we concluded that **2e** induces gene expression mainly in a *JAZ9*-selective manner. **2e** effectively reduces the redundancy of the COI1-JAZ co-receptors and it led to the identification of specific genes most likely regulated by the JAZ9 repressor (Fig. 4b and Supplementary Fig. 18 as a list of genes).

Previous achievements demonstrated that the limitations of the genetic approach, such as redundancy or lethality of essential genes, could be overcome by chemical genetics[48,49]. We hypothesized that transient inhibition/degradation of a single JAZ protein in fully developed plants may disclose the function of a single *JAZ* gene within the redundant *JAZ* gene family. Our results demonstrated the uniqueness of "chemical mutants" in which gene redundancy does not seem to compensate the transiently degraded protein. We showed that this chemical genetic approach is complementary to genetic approaches[25,26]. The current result paves the way for the rational design of JAZ-subtype selective agonists in crops and edible plants, which will elucidate the role of each *JAZ* gene in the highly redundant *JAZ* family of these plants.

## Methods

### Chemical syntheses
Details of chemical syntheses are described in Supporting Information.

### Fluorescence anisotropy measurement
Fluorescence anisotropy titration experiments were performed at 25 °C in 50 µL of Tris buffer (50 mM Tris-HCl, pH 7.8, 100 mM NaCl, 10% Glycerol, 0.1% Tween20, 20 mM 2-mercaptoethanol (2-ME) and 1 µM inositol-1,2,4,5,6-pentakisphosphate, IP$_5$) containing 100 nM COI1 and 100 nM Fl-JAZPs using a 96-well clear-bottom plate unless otherwise noted. Each ligand was added to the solution and incubated at 25 °C until the *r*-value was not-fluctuated, and anisotropy intensities were measured ($\lambda_{ex}/\lambda_{em}$ = 485 nm/516 nm). *r* values were calculated using the following equation: $r = (I_{VV} - G \times I_{VH})/(I_{VV} + 2 G \times I_{VH})$, $I_{VV}$ and $I_{VH}$ are the fluorescence intensities observed through polarizers parallel and perpendicular to the polarization of the exciting light, respectively, and *G* is a correction factor to account for instrumental differences in detecting emitted compounds. Fluorescence anisotropy was recorded on an EnVision 2105 (PerkinElmer, USA).

### Pull-down assay of *AtCOI1-AtJAZ* peptide
Purified GST-COI1 (5 nM) with ASK1 and each compound (5 µM) are dissolved in 350 µL of Tris buffer and added Fl-JAZPs (10 nM). After incubation on ice for 1 h, the mixture was combined with an anti-Fluorescein antibody (GENETEX, GTX26644, 0.2 µL) and incubated at 4 °C for 5 h with rotation. After incubation, the mixture was combined with Protein G beads (Bio-Rad, 1614023, 10 µL in 50% Tris buffer slurry) and incubated at 4 °C for 1 h with rotation. After incubation, the sample was washed five times with PBS-T. The washed beads were resuspended in 35 µL of SDS-PAGE loading buffer containing dithiothreitol (DTT, 100 mM). After heating for 10 min at 60 °C, the samples were subjected to SDS-PAGE and analyzed by western blotting. SDS-PAGE and western blotting were carried out using a Mini-Protean III electrophoresis apparatus (Bio-Rad, Hercules, CA). Chemiluminescence was observed on Amersham Imager 680 (Cytiva, USA).

### Plant materials and growth conditions
*Arabidopsis thaliana* Col-0 is the genetic background of wild-type and mutant lines used in this study. For biological assay using seedlings, surface-sterilized seeds were grown under a 16-h-light (75 µmol m$^{-1}$ s$^{-1}$; cool-white fluorescent light)/8-h-dark cycle at 22 °C on a half-

strength Murashige-Skoog (MS) plate (2% sucrose, 0.4% gellan gum) or liquid medium (0.5% sucrose) in a Biotron LPH-240SP growth chamber (Nippon Medical & Chemical Instruments Co., Ltd., Osaka, Japan) after vernalization in the dark at 4 °C for 2 days. For biological assay using adult plants, seeds were grown directly in soil or 1:1 soil/vermiculite under a 12-h-light (75 µmol m$^{-1}$ s$^{-1}$, cool-white fluorescent light)/12-h-dark cycle at 22 °C.

For the 35S-JAZ1-GFP line, Amplified 35S-JAZ3-GFP from pGWB5-JAZ2[5,50] was inserted into the *Sal*I-*Xho*I site of pET-32a (Invitrogen). pET-32a-35S-JAZ1-GFP was created by an in-fusion reaction between amplified ORF of *JAZ1* and pET-32a-35S-GFP amplified by inverse PCR of pET-32a-35S-JAZ3-GFP. The adapter (*Eco*RI-*Sal*I-*Xho*I) was inserted into the *Hind*III-*Sac*I site of pGWB5-JAZ2. And then, 35S-JAZ1-GFP in prepared pET-32a-35S-JAZ1-GFP was introduced into the *Sal*I-*Xho*I site of the adapted pGWB5. For the 35S-JAZ9-GFP line, amplified *JAZ9* was introduced into pET-32a-35S-GFP by the in-fusion reaction. Amplified 35S-JAZ9-GFP was introduced into the *Sal*I-*Xho*I site of the adapted pGWB5.

Agrobacterium GV3101, containing these constructs, was used to transform Col-0 plants by floral dipping (Clough & Bent, 1998, conducted by INPLANTA INNOVATIONS Inc., Japan). Transgenic lines were checked for Hygromycin resistance, and lines carrying one insertion were driven to homozygous and used for further experiments.

The mutant lines have been previously described: *coi1-1*, *jaz9-1* (SALK_004872), and *jaz10-1* (SAIL_92_D08)[51–54]. The marker lines 35S: JAZ1-GUS, 35S: JAZ9-GUS, and 35S: JAZ10-GUS were previously described[4,55].

### GUS-reporter assay
Four-day-old seedlings on a half-strength MS plate were transferred to a half-strength MS liquid medium in the presence or absence of each compound. After treatment for 2 h, each seedling was fixed with 90% aqueous acetone for 20 min on ice. Each seedling immersed in GUS staining buffer (50 mM phosphate buffer; pH 7.0; 1 mM K$_4$Fe(CN)$_6$; 1 mM K$_3$Fe(CN)$_6$; 0.1% Triton X-100; 2 mM 5-bromo-4-chloro-3-indolyl β-D-glucuronic acid (X-Gluc, Biomedical Science, Japan)). After staining at 37 °C for two days, each seedling was pictured with an E-520 digital camera (Olympus Corp., Japan).

To quantify the GUS activity, 9–11 roots of ligands-treated seedlings were flash-frozen, and homogenized with extraction buffer (50 µL; 50 mM phosphate buffer, pH 7, 10 mM EDTA, 10 mM 2-mercaptoethanol, 0.1% sarcosyl (*N*-lauroylsarcosine sodium salt; >94%, Sigma-Aldrich) and 0.1% Triton X-100). After centrifugation (15,000 × *g*, 4 °C, 10 min), each supernatant was collected. 10 µL extracts were incubated with a 40 µL extraction buffer containing 1 mM 4-MUG (methylumbelliferyl-β-D-glucuronide hydrate; ≥98%, Sigma-Aldrich). 10 µL samples were collected at *t* = 0 and *t* = 1 h. Then the reaction was quenched with 90 µL of 0.2 M Na$_2$CO$_3$. Fluorescence (Ex./Em. = 365/460 nm) was detected using the spectrophotometer Infinite M200Pro (TECAN, Switzerland). Each fluorescence intensity was normalized by the amounts of proteins determined by the Bradford assay (595 nm).

### Chemical treatment for biological assay using adult plants
For growth inhibition assay, 2 µL of an aqueous solution containing each compound were dropped on the rosette center of a leaf of 2-week-old plants. After incubation for 6 days, each compound was applied again similarly. Pictures of plants were taken after 6 days of definitive treatment. The fresh weight of the aerial part was measured, and then metabolite analyses were performed in the same way as seedlings.

For gene expression analysis, leaves of 3-week-old plants were treated with each compound. After incubation for 6 days, each compound was applied again similarly. After treatment for 8 h, each leaf was flash-frozen, and then total RNA extraction and RT-qPCR were performed in the same way as seedlings.

### Root growth inhibition analysis using seedlings
Two-day-old seedlings on half-strength MS plate were transferred on half-strength MS plate in the presence or absence of each compound. After treatment for 4 days, each seedling was pictured with an E-520 digital camera (Olympus Corp., Japan), and the root length of these seedlings was measured using Image J 1.45S software (http://imagej.net/Welcome).

### Fungal infection analyses
Seeds were grown directly in the soil. As previously described, three leaves (leaf numbers 4, 5, and 6) of 3-week-old plants were inoculated with *A. brassicicola* suspension of 5 × 10$^6$ spores/ml PDB (Difco)[56]. Compound was applied twice at 5 ho before pathogen inoculation and at the same time of pathogen inoculation. Pictures of infected plants were taken 5 days after inoculation. Quantification of spores was carried out in a hemocytometer under a light microscope (Leica). Five to eight inoculated leaves of at least six different plants were pooled for each biological sample, and four to six independent biological replicates were measured for each treatment. Results were analyzed by one-way ANOVA/Tukey HSD post hoc test ($p < 0.05$). The infection assay was repeated three times with similar results. Data are shown as mean ± SEM.

### RNAseq analysis
Six-day-old seedlings (5–8 seedlings/sample) grown on half-strength MS liquid medium were treated with half-strength MS liquid medium in the presence or absence of each compound. After treatment for 8 h, each sample was flash-frozen, and then total RNA was extracted using RNeasy Mini Kit (Qiagen

Co. Ltd., Germany). RNAseq libraries were prepared using the low-cost and easy RNA-SEQ (Lasy-Seq) method described previously[57]. Paired-end sequencing was performed using HiSeq X Ten (Illumina, San Diego, CA). RNA-seq analyses were performed with three biological replications, and the data was analyzed using DESeq2.

Low-quality reads and adapter were trimmed using trimmomatic version 0.39 (http://www.usadellab.org/cms/?page=trimmomatic) with 'ILLUMINACLIP: TruSeq3-SE.fa:2:30:10 LEADING:3 TRAILING:3 SLIDINGWINDOW:4:15 MINLEN:36' options. HISAT2 (http://daehwankimlab.github.io/hisat2/) version 2.2.1 was used to align the reads for reference genome (https://phytozome.jgi.doe. gov/pz/portal.html#!info?alias=Org_Athaliana) with '–max-intronlen 5000' option. Aligned reads within gene models were counted using featureCounts version 2.0.1 (http://subread.sourceforge.net/) with '--fracOverlap 0.5 -O -t gene -g ID -s 1 --primary' options. Differentially expressed genes were identified using R version 4.0.4 (https://www.r-project.org/) and DESeq2 version 1.30.1 (https:// bioconductor.org/packages/release/bioc/html/DESeq2.html) package.

Heat map analyses were performed with an online tool heatmapper[58]. The normalized $\log_2$ values were used to compare the transcriptomic changes using MapMan 3.6.0RC1 (Supplementary Fig. 19)[59]. Gene ontology (GO) enrichment analyses were carried out using the TAIR website (https://www.arabidopsis.org/ tools/go_term_enrichment.jsp).

**RT-qPCR of *Arabidopsis thaliana* seedlings.** six-day-old seedlings (5–8 seedlings/ sample) grown on half-strength MS liquid medium were treated with half-strength MS liquid medium in the presence or absence of each compound. After treatment for 2 or 8 h, each sample was flash-frozen, and then total RNA was extracted using ISOGEN (NIPPON GENE, Japan). Then first-strand cDNA was gained with ReverTra Ace reverse transcriptase (Toyobo, Japan) with oligo-dT primers. A StepOnePlus Real-Time PCR System (Life Technologies, USA) was used for quantitative PCR (all primers sequences for RT-qPCR in Supplementary Data 2). Polyubiquitin 10 was used as a reference gene. To select homozygous *coi1-1*, grown 3-day-old heterozygous *coi1-1* on a half-strength MS plate containing 10 μM JA was transferred to a half-strength MS liquid medium. After 3 days of culturing, each seedling was treated with each compound.

**Statistics and reproducibility.** All the experiments were performed three or more times with independent samples and reproduced with similar results. Quantitative values are expressed as the mean values ± S.D. or S.E. Student's t-test was performed between two groups using Microsoft Office Excel. Analysis of variance (ANOVA) was performed among more than two groups using the CoStat (version 6.400) software.

**Reporting summary.** Further information on research design is available in the Nature Portfolio Reporting Summary linked to this article.

## Data availability

The data supporting the findings of this study are available in the paper and its Supplementary Information. All data generated or analyzed are available from the corresponding authors on reasonable request. The raw RNAseq data can be downloaded from the Gene Expression Omnibus repository (GSE184730). All source data to create all bar and box plots in the main text are described in Supplementary Data 3.

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

## Acknowledgements
We thank Dr. Takuya Kaji (Tohoku University) for his support in preparing Figures. We also thank Dr. Yasuhiro Ishimaru, Mr. Wataru Kozaki, and Ms. Hikaru Hoshikawa (Tohoku University) for their technical assistance. The computations in this study were performed using the TSUBAME3.0 supercomputer at Tokyo Institute of Technology, Research Center for Computational Science of Institute for Molecular Science (IMS), and Research Center for Advanced Computing Infrastructure of Japan Advanced Institute of Science and Technology (JAIST). This work was financially supported by a Grant-in-Aid for Scientific Research from JSPS, Japan (nos. 22KK0076, 21K19037, 20H00402, JPJSBP120229905, and JPJSBP120239903 for M.U., nos. 18H02101, 19H05283, and 21H00270 for Y.T., no. 19K06968 for N.K., 19K05378 and 20H04791 for H.S.), SUNBOR GRANT (N.K.), JSPS Core-to-Core Program Asian Chemical Biology Initiative (M.U.), Nagase Science and Technology Foundation (M.U.), and the Spanish Ministry of Science and Innovation AEI/FEDER (no. PID2019-107012RB-I00 for R.S. and A.C.).

## Author contributions
Conceptualization: M.U.; methodology: M.U., K.H., N.K., H.S., K.B., and Y.T.; validation: K.H., Y.T., R.S., M.S., R.S., and M.U.; formal analysis: K.H., H.N., M.N., A.C., H.S., Y.T., K.B., S.T., M.T., and A.N.; investigation: K.H., H.N., M.N., A.C., H.S., Y.T., K.B., S.T., M.T., A.N., R.S., M.S., and M.U.; resources: N.K., K.M., and K.O.; data curation: A.C., Y.T., K.B., A.N., M.S., R.S., and M.U.; writing—original draft preparation: M.U.; writing—review and editing: K.H., N.K., A.C., K.B., M.S., R.S., and M.U.; visualization: K.H., H.N., M.N., A.C., Y.T., and K.B.; supervision: M.U.; project administration: M.U.; funding acquisition: M.U., Y.T., and N.K. All authors have read and agreed to the published version of the manuscript.

## Competing interests
The authors declare no competing interests.
