## [Peer Review File · Communications Biology]

Reviewers' comments:

Reviewer #1 (Remarks to the Author):

The manuscript entitled "Subtype-selective agonists of plant hormone co-receptor COI1-JAZs identified from the stereoisomers of jasmonate mimic" combines genetic and chemical approaches to give a piece of extensive genetic information about the role of the 2e molecule (a coronatine stereoisomer) in planta. 2e would form a specific COI1-JAZ9 co-receptor complex due to the high selectivity affinity that shows this molecule for the co-receptor in fluorescence anisotropy (FA) assays. In contrast to coronatine (2), 2e does not affect growth promoter genes (such as expansin and XTHs) which could explain why coronatine arrests plant growth. Moreover, the authors showed that 2e upregulated defense-related genes of the ERF/ORA branch to a great extent in a JAZ9-dependent manner. This study reveals the function of JAZ9 as a repressor for certain specific genes of JA responses. The entire work is concise and well-documented, and the problem is addressed by different chemical, biochemical, genetic, and molecular approaches that bring interesting biological information about the specific role of JAZ repressors.

I think there are some points to be addressed by the authors:

1.- I noticed that the affinity of 2e to the AtCOI1-AtJAZ9 pair by pull-down assays is not highly selective as the authors found in FA assays according to that they show in two assay replicates (compare images for 2e in Fig. S6B and S6C versus 2e in Fig. S6A where JAZ10 and JAZ12 would show certain affinity to form the complex, see attached ppt). Although the selectivity for JAZ9 in comparison to JAZ10 is clear by GUS activity (Figs. 2B, S8) and genetic analysis (Fig. S20), I don't know in the case of JAZ12.

2.- Is the root system of 2e-treated plants like the mock ones? If possible, authors could show the root growth of 2- and 2e-treated plants to complement the growth inhibition assay (fig. 3A).

3.- Lines 155-156: Why does SIJAZ7 is also excluded of the FA assay? Please add a brief explanation in the ms text.

4.- Is it possible to include inositol pentakisphosphate (InsP5) in the in silico analysis of AtCOI1-2-AtJAZ9 and AtCOI1-2e-AtJAZ9 complexes?

Other points:

-Lines 262-263: "...revealed the unique function of JAZ9, which did not affect the downregulation of genes, including cell-wall-related ones which play important roles in the plant root growth inhibition,". If authors refer to EXLA2, XTH19 and XTH23 genes, these are positive regulators of cell elongation, so I think the above sentence ("which play important roles in the plant root growth inhibition") is incorrect.

-I think a more precise title for this ms should be: "Subtype-selective agonists of plant hormone co-receptor COI1-JAZs identified from the stereoisomers of coronatine"

-Line 125: it should be Fig. S4

-Line 183: change to "Reverse transcription-qPCR (RT-qPCR)" (this is according to the MIQE guidelines by Bustin et al. 2009). Change it through the ms.

-Lines 236, 252: include the full name for MOA, NOPh

-Lines 363-364: enzyme names should be corrected (italics only for initials of genus and species names)

Reviewer #2 (Remarks to the Author):

1. Brief summary of the manuscript

This manuscript examines through chemical genetics the possibility to target individual JAZ genes

functions in Arabidopsis (and as a short extension in) tomato. JAZ are transcriptional repressors in the jasmonate hormonal pathway and are encoded by fairly large gene families, resulting in notoriously high genetic redundancy. This complicates identification of individual JAZ protein functions in mediating JA responses. Whereas coronatine (COR, compound 2), a bacterial mimick of the natural bioactive jasmonate JA-Ile, promotes non-selective assembly of COI1 with most JAZ proteins, the conceptual advance here is that authors generated by combinatorial synthetic chemistry a library of 16 COR stereoisomers and through molecular assays exploration, they uncovered subtype-specific interactions : 1 or 2 isomers (2e, 2f) interacts only with one (AtJAZ9) or a few JAZ. 2e specifically triggers JAZ9 degradation, and this is associated with upregulation of subsectors of defense and enhanced fungal resistance. Global comparative transcriptome analysis revealed the set of defense- and JA-related genes specifically induced via JAZ9 degradation. Contrary to COR effects, this comes along with no downregulation of growth-related genes, which is taken as an explanation why 2e does not inhibit growth, and rules out a role of JAZ9 in JA-mediated growth inhibition.

2. Overall impression of the work

The work is an elegant contribution of chemical genetic approach to overcome genetic redundancy in the functional analysis of COI1-JAZ co-receptors. It first highlights that co-evolution of plant-Pseudomonas interactions resulted in a natural stereoisomer of coronatine (2) that non-selectively targets most JAZ subtypes for degradation, and hence maximizes induction of JA responses as a decoy defense. The combinatorial chemistry used here creates structural variability in which some stereochemical modifications disrupt totally formation of COI-JAZ co-receptors, or largely unaffected these latter. The smartness of the study is to have exploited the rare occurrences where a stereoisomer specifically retains interaction with one (2e) or a few (2f) JAZ isoforms.

The manuscript is nicely written and integrates a globally convincing set of experimental approaches allowing to conclude as to JAZ9-specific functions. The molecular, biochemical and physiological data on 2e triggering specifically JAZ9 degradation and responses are well supported by refined transcriptome analysis and discussion. The differential effects of 2 and 2e on growth inhibition and distinct transcriptional changes provide novel and useful insights into JAZ9 function in JA responses.

One point that should certainly be made more clear relates to the distinct consequences of JAZ9 genetic ko and 'protein ko' through 2e action, which are well illustrated. However, this result is not really explained/discussed : why can JAZ9 deficiency be complemented in jaz9-1 by redundancy but not after 2e action in WT ? Other JAZ need a 'generic' ligand, such as the endogenous JA-Ile, but this should work in both WT and jaz9-1. Is this a matter of dynamic ? rapid upon 2e treatment, more slow adaptation in jaz9-1 ?

3. Specific comments, with recommendations for addressing each comment

From Fig. S4, it seems that 2e is specific but not very potent

Fig. S5: would be easier to read if layout of Fig. S5 was the same as Fig. 2A (Kd)

Figure 3a and S12: pictures appear too dark

L177: needs re-phrasing : Fig. 3C and 3D do not show defenses, but symptoms. How was this treatment realized ? How long before inoculation were plants treated ? Compound 2 is said to strongly inhibit growth, but here leaf size is the same as mock.

L177-79: 'in which other members...of JAZ9 protein' : to make this comment more clear, one should add 'of JAZ9 protein in presence of non-selective ligand'.

L182: again, 'a defense response' is not shown, but a macroscopic phenotype

L183 and legend Figure 3F : it should be made clearer if gene expression is in inoculated or non-inoculated plants.

L. 186:

Figure 3C : spores measurement is nice, but picture is difficult to assess as to leaf colonization. Relative fungal biomass determined by qPCR would be more reflective.

L227-28: the sentence 'The unique downregulated gene expression...' comes too early or is redundant with the last phrase of the paragraph, should be more hypothetical at first.

L236: MOA: not defined.

L211 : how is difference between genetic and chemical explained ?

L250: sporulation is a quite distant readout of a defense response. Better say 'resistance response' ?

L254: is this correct ? 2e DOES induce the ORF/ORA branch in jaz10-1 like in WT.

L260: activation ?

L307: low but passes threshold...

L411: calls for Botrytis inoculation but Alternaria was shown..

Fig S20A: too dark

Flesh weight versus fresh weight: homogenize

Some typos to correct.

Point-by-Point Responses to Reviewers' comments

We thank the reviewers for their encouraging comments and constructive suggestions and have responded to their specific comments on each point and concern below.

Reviewer #1 (Remarks to the Author):

The manuscript entitled "Subtype-selective agonists of plant hormone co-receptor COI1-JAZs identified from the stereoisomers of jasmonate mimic" combines genetic and chemical approaches to give a piece of extensive genetic information about the role of the 2e molecule (a coronatine stereoisomer) in planta. 2e would form a specific COI1-JAZ9 co-receptor complex due to the high selectivity affinity that shows this molecule for the co-receptor in fluorescence anisotropy (FA) assays. In contrast to coronatine (2), 2e does not affect growth promoter genes (such as expansin and XTHs) which could explain why coronatine arrests plant growth. Moreover, the authors showed that 2e upregulated defense-related genes of the ERF/ORF branch to a great extent in a JAZ9-dependent manner. This study reveals the function of JAZ9 as a repressor for certain specific genes of JA responses. The entire work is concise and well-documented, and the problem is addressed by different chemical, biochemical, genetic, and molecular approaches that bring interesting biological information about the specific role of JAZ repressors.

I think there are some points to be addressed by the authors:

1.- I noticed that the affinity of 2e to the AtCOI1-AtJAZ9 pair by pull-down assays is not highly selective as the authors found in FA assays according to that they show in two assay replicates (compare images for 2e in Fig. S6B and S6C versus 2e in Fig. S6A where JAZ10 and JAZ12 would show certain affinity to form the complex, see attached ppt). Although the selectivity for JAZ9 in comparison to JAZ10 is clear by GUS activity (Figs. 2B, S8) and genetic analysis (Fig. S20), I don't know in the case of JAZ12.

Ans.

Thank you for your critical comment. Although the affinity of 2e with JAZ12 seems very weak, we also examined GUS activity (Supplementary Fig.8), the effect on growth inhibition (Supplementary Fig.23A&B), and genetic analysis on *the jaz12* mutant line (Supplementary Fig.23C). The result showed the selectivity for JAZ9 in comparison to JAZ12. However, the GUS-staining of JAZ12-GUS was weak compared to those of other JAZ-GUS, thus, we added only the bar chart representation for JAZ12-GUS in Supplementary Fig. 8. Therefore, we revised the sentence in line 137-139 as follows: "*In planta* AtJAZ9-selectivity of 2e was tested by using a GUS reporter assay using *A. thaliana* reporter lines, 35S: JAZ1-GUS, 35S: JAZ9-GUS, 35S: JAZ10-GUS, and 35S: JAZ12-GUS (Fig. 2B and Supplementary Fig. 8). All

JAZ-GUS”

And we revised Figure legend of Supplementary Fig.8: Evaluation of GUS activity in the roots of 4-day-old 35S: *JAZ1-GUS*, 35S: *JAZ9-GUS*, 35S: *JAZ10-GUS*, and 35S: *JAZ12-GUS* seedlings (n=4).

In addition, we added the following sentence in line 256-257: “Moreover, no growth inhibition and expressions of marker genes belonging to ERF/ORF branch were observed in 2e-treated *jaz12-1* (Supplementary Fig. 23).”

2.- Is the root system of 2e-treated plants like the mock ones? If possible, authors could show the root growth of 2- and 2e-treated plants to complement the growth inhibition assay (fig. 3A).

Ans.

Thank you for your critical comment. We added the root growth inhibition assay using *Arabidopsis* seedlings as Supplementary Fig. 13. COR (2) induced severe growth inhibition in *Arabidopsis* roots, while 2e did not affect the root system like the mock-treated plants. These results showed that 2e did not affect the root growth in *Arabidopsis*, similar to the above-ground part. Therefore, we added the sentence in lines 172-173: "In addition, no growth inhibition was observed on root by the addition of 2e (Supplementary Fig. 13).”

According to this revision, we added the corresponding method in lines 413-417 of the ‘Method’ section as follows: ‘Root growth inhibition analysis using seedlings Two-day-old seedlings on half-strength MS plate were transferred on half-strength MS plate in the presence or absence of each compound. After treatment for 4 days, each seedling was pictured with an E-520 digital camera (Olympus Corp., Japan), and the root length of these seedlings was measured using Image J 1.45S software (<http://imagej.net/Welcome>).’

3.- Lines 155-156: Why does SIJAZ7 is also excluded of the FA assay? Please add a brief explanation in the ms text.

Ans.

Thank you for your important comment. As shown in the following Figure, we excluded SIJAZ7 in the FA assay because the non-specific binding of the SICO11-SIJAZ7P pair was observed in the absence of a ligand. This was described in Figure legend of Fig. 2C as “Unfortunately, we cannot examine the affinity of stereoisomers with the SICO11-SIJAZ7P pair in the FA assay because the severe non-specific binding was observed in the absence of ligand”. Thus, we moved this sentence to lines 156-158 of the main text.

4.- Is it possible to include inositol pentakisphosphate (InsP5) in the in silico analysis of AtCOI1-2-AtJAZ9 and AtCOI1-2e-AtJAZ9 complexes?

Ans.

Thank you for your critical comment. The crystal structure of AtCOI1-2-AtJAZ1 (PDB 3OGM) includes PO4 instead of InsP5, and previous reports showed that inositol oktakisphosphate (InsP8) also form stable complex COI1-2-JAZ instead of PO4. (M. Cui et al., *Front. Plant Sci.*, 2018; M. Cui et al., *Comput. Aided Mol. Des.*, 2022) We re-performed docking including InsP8 and following MD simulation to obtain binding models AtCOI1-2-InsP8-AtJAZ9 and AtCOI1-2e-InsP8-AtJAZ9. The result showed that 2 and 2e also accommodated in the binding pocket of AtCOI1-AtJAZ9. Therefore, supplementary Fig. 7 was replaced by the results of MD simulation of AtCOI1-2-InsP8-AtJAZ9 and AtCOI1-2e-InsP8-AtJAZ9 including InsP8, and accordingly, the sentence in Lines 132-137 was rephrased as follows: ‘Docking and subsequent in silico molecular dynamics (MD) simulations demonstrated that 2e, as well as 2, can be accommodated in the binding pocket of COI1-InsP8-JAZ9 complex (Supplementary Fig. 7). In this MD simulation, we used inositol oktakisphosphate (InsP8) instead of inositol pentakisphosphate (InsP5) because previous reports showed that InsP8 also formed stable complex with COI1-2-JAZ instead of phosphates (PO4).^{37,38.}’

According to this revision, we revised the corresponding method in supplementary methods section as follows: ‘The initial structure of the COI1-2-JAZ1 complex was obtained from the crystal structure (PDB ID: 3OGM). COI1-2-JAZ9 complex was obtained according to our previous report.⁴ In order to model the complex structure of the COI1-2-InsP8-JAZ9, the structures of inositol phosphates (PO4) in the COI1-2-JAZ9 complex were removed, and then the InsP8 structure extracted from the other protein structure (PDB ID: 3T9F) was

docked to the removed sites of PO4. The structure of COI1-2e-InsP8-JAZ9 was prepared by replacing 2 with 2e by docking simulation. DOCK 6.6 software was used for the docking of the InsP8 and 2e, and Amber99 force field parameters were assigned for the estimations of grid score. The space of the conformation search was defined in a 12 Å radius from the center of the binding sites of COI1. The best score pose was used for the subsequent MD simulations in a water solvent. Five independent 100 ns MD simulations with different initial velocities were performed to sample the equilibrated structures of COI1-2-InsP8-JAZ9 and COI1-2e-InsP8-JAZ9. A Parrinello-Rahman type thermostat⁵ and a Nosé-Hoover type barostat⁶ were adopted to control the system temperature (T = 300 K) and pressure (P = 1 atm). The force field parameters of CHARMM36⁷, generalized charmm force field⁸, and TIP3P water model⁹ were assigned for the protein, ligand, and water molecule, respectively. The cutoff length for the actual space was set to 12 Å. The Particle mesh Ewald (PME) method¹⁰ was used to treat long-range electrostatics. The integration time step of the system was 2 fs, and the MD structures were stored every 10 ps. All MD calculations were done using the GROMACS 2018 program package. Radial distribution functions (RDFs) were calculated to analyze the hydrogen bond networks between the compounds and surrounding residues in the binding pocket of COI1. The RDF curves for possible hydrogen bond pairs were evaluated using a total 500 ns term MD simulation. All MD trajectory data (total 500 ns) were used for the RDF analysis.'

Other points:

-Lines 262-263: "...revealed the unique function of JAZ9, which did not affect the downregulation of genes, including cell-wall-related ones which play important roles in the plant root growth inhibition,". If authors refer to EXLA2, XTH19 and XTH23 genes, these are positive regulators of cell elongation, so I think the above sentence ("which play important roles in the plant root growth inhibition") is incorrect.

Ans.

Thank you for your important comment. We rephrased the sentence in lines 263-265 as "...revealed the unique function of JAZ9, which did not affect the downregulation of genes, including cell-wall-related ones which play important roles in the plant root growth,".

-I think a more precise title for this ms should be: "Subtype-selective agonists of plant hormone co-receptor COI1-JAZs identified from the stereoisomers of coronatine"

Ans.

Thank you very much for your valuable comment.

We corrected the title as you suggested.

-Line 125: it should be Fig. S4

Ans.

Thank you for your kind remark.

I corrected the figure number in line 124.

-Line 183: change to “Reverse transcription-qPCR (RT-qPCR)” (this is according to the MIQE guidelines by Bustin et al. 2009). Change it through the ms.

Ans.

Thank you for your kind remark.

I have corrected the description in lines 182-3, 228, 244, 410-11, 455, 462, 701 and 706 in the legend of Figure 4, and the legend of Supplementary Figures 17, 21, and 22, as you indicated.

-Lines 236, 252: include the full name for MOA, NOPh

Ans.

Thank you for your kind remark.

I spelled them out as follows:

MOA is ‘mode of action’ in Line 236

NOPh is ‘*N*-coronafacyl *ent*-coronamic acid *O*-phenyloxime (NOPh)’ in Line 252-3

In addition, ‘*ent*-CFA-CMA-*O*-phenyloxime (NOPh)’ was corrected to NOPh.

-Lines 363-364: enzyme names should be corrected (italics only for initials of genus and species names)

Ans.

Thank you for your kind remark. We corrected the “JAZ1” in line 366 and “JAZ9” in line 370 as italics and corrected the restricted enzyme names in lines 365, 367, 368, 369, and 371-372.

Reviewer #2 (Remarks to the Author):

One point that should certainly be made more clear relates to the distinct consequences of JAZ9 genetic ko and 'protein ko' through 2e action, which are well illustrated. However, this result is not really explained/discussed : why can JAZ9 deficiency be complemented in jaz9-1 by redundancy but not after 2e action in WT ? Other JAZ need a 'generic' ligand, such as the endogenous JA-Ile, but this should work in both WT and jaz9-1. Is this a matter of dynamic ? rapid upon 2e treatment, more slow adaptation in jaz9-1 ?

Ans.

Thank you for your important comment. Each JAZ gene expresses at proper time and location in the plant body to adapt the environmental changes. In jaz KO mutant lines, the spatiotemporal expression of other members of the jaz family gene could change during the process of growth and development, and this change could compensate for the function of the knocked-out jaz gene. Thus, we hypothesized that this complementation might not occur in fully developed plants. Therefore, we added the sentence in Lines 316-318 as follows: 'We hypothesized that transient inhibition/degradation of a single JAZ protein in fully developed plants may disclose the function of a single JAZ gene within the redundant JAZ gene family.'

3. Specific comments, with recommendations for addressing each comment

From Fig. S4, it seems that 2e is specific but not very potent

Ans.

Thank you for your helpful comment. As you pointed out, the bioactivity of 2e was moderate. However, throughout our current results, 2e was used as a valuable tool for the dissection of complex jasmonate signaling. Thus, we rephrased the wording of line 33 from 'potent' to 'valuable' according to your suggestion.

Fig. S5: would be easier to read if layout of Fig. S5 was the same as Fig. 2A (Kd)

Ans.

Thank you for your kind suggestion. We revised the layout of Supplementary Fig. 5, the same as Fig. 2A, and that of Supplementary Fig. 11 same as Fig. 2C.

Figure 3a and S12: pictures appear too dark

Ans.

Thank you for your kind comment. We improved the contrast of indicated pictures.

L177: needs re-phrasing : Fig. 3C and 3D do not show defenses, but symptoms. How was this treatment realized ? How long before inoculation were plants treated ? Compound 2 is said to

strongly inhibit growth, but here leaf size is the same as mock.

Ans.

Thank you for your important comments about the bacterial infection experiment. We corrected the sentence in lines 173-176 from 'However, 2e, as well as 2, moderately activated defenses against the infection of fungal pathogen *Alternaria brassicicola* compared to the mock-treatment (Fig. 3C and 3D)' to 'However, 2e, as well as 2, moderately activated resistance response against the infection of fungal pathogen *Alternaria brassicicola* compared to the mock-treatment (Fig. 3C, 3D, and Supplementary Fig. 14)'.

In lines 421-423 of the Methods section, treatment procedures were added as "Compound was treated twice at 5 hours before pathogen inoculation and at the same time of pathogen inoculation."

The different chemical treatment conditions because of the assay requirement explain the different effects on growth inhibition. The compounds must be administered during the germination-to-growth period to see their effects on growth inhibition. For this reason, in the growth inhibition analysis (Figure 3A), treatments by 2 and 2e were repeated until the seedlings grew to 2-week-old. On the other hand, fungal infection assays should be performed on fully mature plants. This is because leaves above a specific size are required to perform inoculation. Thus, in fungal infection analysis (Figure 3C), 2 and 2e were treated on the adequately developed adult plants. In this condition, plants were fully grown, and thus, no growth inhibition occurred.

L177-79: 'in which other members...of JAZ9 protein' : to make this comment more clear, one should add 'of JAZ9 protein in presence of non-selective ligand'.

Ans.

Thank you very much for your kind comment.

We rephrased the sentence in Lines 176-178 as 'in which other members of the JAZ family protein likely complemented the original function of JAZ9 protein in presence of non-selective ligand,'

L182: again, 'a defense response' is not shown, but a macroscopic phenotype

Ans.

Thank you very much for your critical comment.

I rephrased 'defense response' to 'resistance response' in Lines 181-182, 250, and 302.

L183 and legend Figure 3F : it should be made clearer if gene expression is in inoculated or non-inoculated plants.

Ans.

Thank you for your important comment. To clarify the treatment condition, we added

'(without pathogen infection)' in line 184 and 'without pathogen infection' in line 685 of figure legend 3F.

L. 186:

Figure 3C : spores measurement is nice, but picture is difficult to assess as to leaf colonization. Relative fungal biomass determined by qPCR would be more reflective.

Ans.

Thank you for your important comment. The result of RT-qPCR analysis of an *Alternaria brassicicola* CUT (AbCUT) gene was added as supplementary Fig. 14 and referred to the sentence in line 173-176 'However, 2e, as well as 2, moderately activated resistance response against the infection of fungal pathogen *Alternaria brassicicola* compared to the mock-treatment (Fig. 3C, 3D, and Supplementary Fig. 14).'

L227-28: the sentence 'The unique downregulated gene expression...' comes too early or is redundant with the last phrase of the paragraph, should be more hypothetical at first.

Ans.

Thank you for your helpful comment. The sentence, 'The unique downregulated gene expression profile caused by 2e explained why 2e did not affect the plant growth.' was removed because it is redundant with the last phrase of the paragraph in lines 232-233, 'The reason why 2e did not cause growth inhibition could be explained by the lack of gene downregulation of these cell-wall-related genes,' as you mentioned.

L236: MOA: not defined.

Ans.

Thank you for your kind remark.

MOA is spelled out as 'mode of action' in Line 236

L211 : how is difference between genetic and chemical explained ?

Ans.

Thank you for your important comment. Each *JAZ* gene expresses at the proper time and location in the plant body to adapt the environmental changes. In *jaz* KO mutant lines, the spatiotemporal expression of other members of the *jaz* family gene could change during the process of growth and development, and this change could compensate for the function of the knocked-out *jaz* gene. Thus, we hypothesized that this complementation might not occur in fully developed plants. Therefore, we added the sentence in Lines 316-318 as follows: 'We hypothesized that transient inhibition/degradation of a single JAZ protein in fully developed plants may disclose the function of a single *JAZ* gene within the redundant *JAZ* gene family.'

L250: sporulation is a quite distant readout of a defense response. Better say 'resistance response' ?

Ans.

Thank you very much for your important comment.

I rephrased 'defense response' to 'resistance response' in Lines 181-182, 250, and 302.

L254: is this correct ? 2e DOES induce the ORF/ORA branch in *jaz10-1* like in WT.

Ans.

Thank you for your kind comment. We rephrased the sentence in lines 254-256 from '2e did not affect the phenotypes and gene expressions in *jaz10-1* mutant: 2e did not cause growth inhibition and up-regulation of defense-related genes belonging to the ERF/ORA branch (Fig. S20).' to '2e did not affect the growth and upregulated defense-related genes belonging to the ERF/ORA branch in *the jaz10-1* mutant (Supplementary Fig. 22).' to avoid confusion.

L260: activation ?

Ans.

Thank you for your kind remark. "de-repression" in line 261 was rephrased as "activation".

L307: low but passes threshold...

Ans.

Thank you for your important comment. JAZ9-independent JA-responsive genes activated by 2e-treatment would be derived from the weak interaction between 2e and some COI1-JAZs except for COI1-JAZ9 in pull-down assay (Supplementary Fig. 6B, C). However, growth and gene expression analyses of 2e-treated *jaz10-1* and *jaz12-1* plants showed that 2e did not affect the growth and upregulated defense-related genes belonging to ERF/ORA branch (Supplementary Fig. 22 and 23). Therefore, RNAseq analysis suggested that 2e mainly activated JAZ9-dependent gene expression belonging to ERF/ORA branch.

L411: calls for Botrytis inoculation but Alternaria was shown..

Ans.

Thank you for the important comment. The fungal infection assay was performed using Alternaria. We corrected the name for bacterial pathogen in line 421.

Fig S20A: too dark

Ans.

Thank you for your kind comment. We improved the contrast of indicated pictures (Supplementary Fig. 22A in this revision).

Flesh weight versus fresh weight: homogenize

Ans.

Thank you for your kind remark. We rephrased “flesh” in lines 674 of Fig. 3B legend and Supplementary Fig. 22 as “fresh.”

Some typos to correct

Ans.

We carefully checked and corrected typos.

REVIEWERS' COMMENTS:

Reviewer #1 (Remarks to the Author):

Thank you for the extensive review of this new ms version. I agree with all changes. Only I have noticed some details in the writing to be reviewed by the authors:

-In the Suppl. Fig. 8 values of JAZ12 lack significant differences (letters?)

-Lines 365-372: Please change the enzyme names to: *Sal*-*Xho*I; *EcoR*-*Sal*-*Xho*I; *Hin*dIII-*Sac*I; *Sal*-*Xho*I; *Sal*-*Xho*I

Reviewer #2 (Remarks to the Author):

The authors have answered properly to my main concerns.

There remain a couple of errors that need to be corrected :

Supp Fig. 14 : if gDNA was extracted, then it should be PCR, not RT-qPCR. As well, the legend of this figure calls for bacteria-infected leaves, but here a fungus is used.

L422: applied twice ?

Point-by-Point Responses to Reviewers' comments

We thank the reviewers for their encouraging comments and constructive suggestions and have responded to their specific comments on each point and concern below.

REVIEWERS' COMMENTS:

Reviewer #1 (Remarks to the Author):

Thank you for the extensive review of this new ms version. I agree with all changes. Only I have noticed some details in the writing to be reviewed by the authors:

-In the Suppl. Fig. 8 values of JAZ12 lack significant differences (letters?)

Ans:

Thank you for your kind suggestion. We added the significant differences in values of JAZ12 in Supplementary Fig. 8.

-Lines 365-372: Please change the enzyme names to: *Sall-XhoI*; *EcoRI-Sall-XhoI*; *HindIII-SacI*; *Sall-XhoI*; *Sall-XhoI*

Ans:

Thank you for your kind corrections. As you suggested, we corrected the restricted enzyme names in lines 365, 367, 368, 369, and 371-372.

Reviewer #2 (Remarks to the Author):

The authors have answered properly to my main concerns. There remain a couple of errors that need to be corrected :

Supp Fig. 14 : if gDNA was extracted, then it should be PCR, not RT-qPCR. As well, the legend of this figure calls for bacteria-infected leaves, but here a fungus is used.

Ans:

Thank you for your kind suggestion. We corrected the sentence in the legend of Supplementary Fig. 14 from "RT-qPCR analysis was performed by using *A. brassicicola* genomic DNA extracted from bacteria-infected leaves shown in Fig. 3C." to "Quantitative PCR analysis was performed by using *A. brassicicola* genomic DNA extracted from fungus-infected leaves shown in Fig. 3C."

L422: applied twice ?

Ans:

Thank you for your kind correction. "treated" in line 422 was rephrased as "applied."